# A Defense of One-Step Learning: Examining Single-Batch Distillations

## Abstract

Dataset distillation produces a compressed synthetic dataset that approximates a large dataset or other learning task. A model can be trained on a distillation in a single gradient descent step. Conventional wisdom suggests that single-step learning is not generalizable and should yield poor performance; yet, distillation defies these expectations with good approximations of direct-task training for a large distribution of models. In order to understand how distilled datasets can perform one-shot learning, we examine the distilled data instances and the cost surfaces produced by the distilled datasets. We demonstrate that the distilled dataset not only mimics features of the true dataset but also produces cost surfaces such that one-step training leads models from the initialization space into local minima of the true task's cost surface. This shows how one-step learning's counter-intuitive success is not only reasonable, but the expected outcome of dataset distillation.

## 1 Introduction

What makes a good dataset? For machine learning, the distribution of data and features in a dataset should approximate those of the underlying real-world distribution. Thus, datasets are generally produced by sampling from the real-world distribution. While an effective solution, a large number of samples must be taken to provide an adequate sampling of the real distribution. For example, the training set of MNIST, often considered a toy problem in image recognition, contains 60,000 examples. Large datasets are information-sparse, which makes learning costly. This cost can be reduced by compressing the dataset into an information-dense representation using dataset distillation.

Dataset distillation produces a small synthetic dataset that approximates a larger learning task. The synthetic dataset can be highly compressed and learnable in a single step of gradient descent. Training a predictive model on the synthetic dataset approximates training the model on the large dataset—the distillation-trained model should perform comparably to the model trained on the original task.

Dataset distillation was introduced by Wang et al. (2018), and similar methods using generative models were introduced by Such et al. (2020) and Huang & Zhang (2021). While both use a meta-learning approach, other learning methods such as gradient matching (Zhao et al., 2021), trajectory matching (Cazenavette et al., 2022), distribution matching (Zhao & Bilen, 2023), differentiable Siamese augmentation (Zhao & Bilen, 2021), and kernel ridge-regression (Nguyen et al., 2021) have been proposed. In this work, we focus on the original meta-learning formulation due to its independence from expert models and trajectories, as well as its simplicity. Using learned soft labels with the original meta-learning approach improves performance (Sucholutsky & Schonlau, 2021b) and allows for the highest-possible compression (Sucholutsky & Schonlau, 2021a). Use cases of distillation include dataset compression, neural architecture search acceleration (Such et al., 2020), data poisoning attacks (Wang et al., 2018), and data anonymity (Dong et al., 2022).

Dataset distillation can provide extraordinary compression—reducing the 60,000 instance MNIST task to 6 instances, and reducing the reinforcement learning *Centipede* environment to 10 instances. It would seem that compressing a dataset into a representation 10,000 times smaller, resulting in fewer instances than the number of classes, should be able to yield only the coarsest approximation of the original training dataset, especially when learned in a single large gradient descent step. However, experimentation proves that distillation can provide close approximation to training on the real task. This begs the question: what is the distillation actually learning and how can such results be

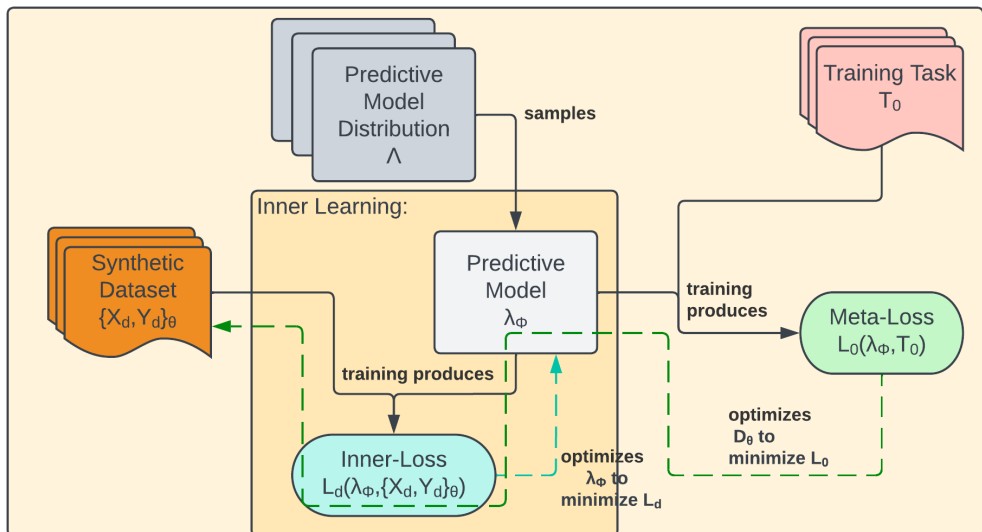

Figure 1: Task-agnostic distillation meta-learning. This process produces a parameterized distiller such that predictive models sampled from $\Lambda$ maximize performance on the original training task $T_0$ when trained on the distilled dataset $\{X_d, Y_d\}_\theta$. This is achieved through a nested learning process. At each outer iteration, the distiller $D_\theta$ produces a synthetic task. A model $\lambda_\phi$ is sampled and initialized from the architecture and initialization distribution $\Lambda$. The inner learning begins, in which $\lambda_\phi$ is trained on the synthetic dataset for a single step. Once trained on the synthetic task, $\lambda_\phi$ is tested against the original training task and a metaloss is produced. The loss is backpropagated through the inner learning to update the distilled dataset's parameters $\theta$. Once the distiller converges, any model from $\Lambda$ can be trained on the synthetic data in a single step.

achieved in a single gradient descent step? We seek to answer these questions by examining distilled datasets using two main techniques: an examination of the distilled instances and an examination of the optimization structure utilized in learning, referred to as the cost surface or loss landscape. We demonstrate that some task-specific interpretability can be provided by the instances themselves, especially in distillations of simpler tasks. We also demonstrate that an examination of the cost surfaces of learner models is vital to understanding how distillation in general can model a large learning task using few distilled instances and one-step learning. As a result, this paper makes the following contributions:

1. Demonstrates interpretability of a task through examining distilled data instances and tasks involving $c$ classes distilled into fewer than $c$ training instances.

2. Demonstrates that distillation can model an original task by mimicking the positions of minima on the cost surface.

3. Demonstrates that distilled datasets can approximate cost surfaces for multiple tasks at significantly lower computational costs.

4. Demonstrates that distillation can produce good approximate cost surfaces on out-of-distribution architectures not seen in distillation training.

5. Demonstrates that synthetic datasets distilled for single-step learning can train a model without cost surface mimicking, revealing cost surface features necessary for learning.

## 2 DISTILLED DATASETS

In this work, we utilize minimum-sized distillations of four tasks: MNIST, CIFAR-10, cart-pole, and Atari *Centipede*. While the focus of this work is not on the production of distillations, we provide the

---

**Algorithm 1:** Task-Agnostic Distillation for One-Step Learning($\{X_d, Y_d\}_\theta$, $\Lambda$, $V_\psi$, $E_0$, $e$, $n$, $b$)

---

**input** : initialized synthetic dataset $\{X_d, Y_d\}_\theta$, learner distribution $\Lambda$, targeted training task $T_0$
**output:** learned synthetic dataset $\{X_d, Y_d\}_\theta$ distilled from $E_0$

**1 while** $\{X_d, Y_d\}_\theta$ *has not converged* **do**
**2**     $\lambda_\phi := \text{Sample}(\Lambda)$;
**3**     $L_{inner} := \frac{1}{n} \sum (\lambda_\phi(X_d) - Y_d)^2$;
**4**     $\nabla_\phi := \text{Backpropagate}(L_{inner}, \phi)$ ;
**5**     Optimize $\phi$ w.r.t. $\nabla \phi$ using differentiable optimization;
**6**     $L := \text{OuterLoss}(\lambda_\phi, T_0)$;
**7**     $\nabla \theta := \text{BackpropagateWithMetaGradients}(L, \theta, \nabla_\phi)$;
**8**     Optimize $\theta$ w.r.t. $\nabla \theta$ ;
**9 end**
**10** return $\{X_d, Y_d\}_\theta$;

---

following details to aid in reproduction, in addition to providing the experiment code and distilled parameters in the supplementary material (code repository will be provided in deanonymized version). MNIST distillation was performed following the procedures of Wang et al. (2018), with added soft-labels (Sucholutsky & Schonlau, 2021b) allowing for a minimum distillation of 6 synthetic instances to be learned with a single step of gradient descent. The combination of single-step learning with minimum-sized distillations significantly reduces the cost of the meta-learning process.

### 2.1 PRODUCING DISTILLATIONS

Distillation is performed through a meta-learning process that creates a dataset such that: 1) it contains compressed information from a targeted training task in its synthetic data instances, and 2) a model trained on the distilled data approximates the performance of a model trained on the targeted training task. Compression and approximation are achieved during the distillation training process.

A visualization of task-agnostic distillation is provided in Figure 1 and the algorithm is provided as Algorithm 1. Since we use a single inner learning step in distillation training, the distiller is trained such that a predictive model will be fully trained on the distilled dataset in a single step of SGD. In our experiments, the distilled dataset consists of a parameterized data instance tensor, a parameterized label tensor consisting of soft label scores rather than hard class labels, and an inner learning rate parameter. Since the synthetic task uses soft labels, we use mean squared error loss as the inner loss function. The meta-loss function depends on the training task: for MNIST and CIFAR-10 we use cross entropy loss, and for the RL environments we use PPO (Schulman et al., 2017).

Using PPO as an outer objective requires the addition of an auxiliary critic model trained alongside the distiller (rather than being reinitialized each outer iteration with the inner learner). Just like in standard PPO, the critic is only used in RL training, not in evaluation nor in training a model on the distilled data. In addition, the RL data gathering is removed from the inner-outer learning structure and the learner is reverted to the same initialization over a set of PPO policy epochs to allow for the trust region protection of PPO to respond to changes in the distiller that occur throughout learning.

### 2.2 DISTILLATIONS USED IN EXPERIMENTS

To represent distillations of supervised classification, we use MNIST and CIFAR-10 training datasets. They both contain 10 image classes: handwritten digits and real-world objects, respectively. MNIST contains 60,000 training instances, and CIFAR-10 contains 50,000. Both are distilled to the minimum-sized distillation of 6 instances (due to having 10 classes (Sucholutsky & Schonlau, 2021a)) to be learned in one gradient descent step. The synthetic data instances of both tasks match the shape of the respective dataset's instances: 1x28x28 for MNIST, 3x32x32 for CIFAR-10. We use a small convolutional network with two convolution layers and a linear output layer, with ReLU activations. The convolutional kernels are 3x3; the first layer has 16 filters, the second has

32. Parameters are initialized using Xavier uniform initialization (Glorot & Bengio, 2010). 1000 models sampled from this distribution achieve an average validation accuracy on MNIST of 84.4% when trained on the distilled dataset for one step, compared to one model that reached 98.5% when trained to convergence on the MNIST training set. On CIFAR-10, 1000 models achieved 28.1% accuracy on the validation set, while one model trained directly on CIFAR-10 reached 56.8% validation accuracy (despite near-perfect training set accuracy). While the CIFAR-10 distillation did not converge well, perhaps due to the model's risk of overfitting, the results demonstrate that even poor distillations function similarly to well-converged ones.

We utilize Gymnasium's environments(Brockman et al., 2016) for cart-pole and *Centipede*, in addition to producing a multi-dimensional extension of cart-pole that vectorizes states and actions into $N$ dimensions. These are distilled into single-batch supervised learning datasets that are learned in a single step, containing instances that match the state size: $4N$ for ND cart-pole and $4 \times 84 \times 84$ for *Centipede*. Cart-pole is distilled into $N+1$ instances and *Centipede* is distilled into 10 instances, due to their action spaces of $2N$ and 16 distinct action classes, respectively. Determining the degree of compression is less straightforward than supervised learning, as RL dynamically creates experience datasets. However, our PPO trial of *Centipede* was learned on approximately $8,000,000$ datapoints over $15,625$ training steps, versus 1 training step on the distilled dataset's 10 instances. When tested in the *Centipede* environment, 1000 randomly initialized models achieve an average reward of 8083 when trained for one step on our distillation as opposed to a model trained on *Centipede* with PPO that reached 9167 reward over 1000 trials. For standard 1D cart-pole, 1000 learners all achieved the maximum reward of 500 in all instances, matching the performance of models trained directly on cart-pole. For 2D cart-pole, the distillation-trained learners achieved an average reward of 364.1, while the models trained directly on 2D cart-pole achieved an average reward of 498.2. We followed the practices and architectures of Huang et al. (2022) for these environments, except for value loss clipping and learning rate annealing. We used PPO as the outer objective with 4 policy epochs, a batch size of 512, 10 episodes per policy epoch, 10 parallel environments, and 200 instances per environment per episode.

## 3    RESULTS

In order to understand distillation and provide task-specific explainability, we examine the data instances and the cost surfaces produced by the distillations described in Section 2.2.

### 3.1    EXAMINING DISTILLED DATA INSTANCES

Distilled dataset instances can be used as interpretability artifacts; however, the interpretability is limited Wang et al. (2018). With minimum-sized distillations, the learned features should all be present in the few data instances; however, high levels of compression can obscure these features to human observers. In simple environments, like cart-pole and its multidimensional extension, the resulting minimum-sized distillation is clearly interpretable: the agent should move the cart towards the direction the pole is leaning (see Figure 2). For larger-dimensional tasks, such as image recognition tasks, the resulting distillation may have some identifiable features of the labeled classes, but are not clearly interpretable (see Figure 3a). For more complicated tasks, such as distillations of Atari *Centipede*, the distillation may appear to be random noise to human observers, despite containing information that can be used by a model to learn the task. There may be ideal hyperparameters or other techniques that can optimize interpretability of a distillation; however, it is well accepted that distilled instances do not need to resemble real data instances, and that forcing resemblance may require increased computational costs or may decrease the effectiveness of the distillation (Wang et al., 2018).

Examining individual instances provides insight into how synthetic soft labels allow a model to be trained to distinguish $c$ classes (or actions) in fewer than $c$ training instances. For 2D cart-pole in Figure 2, each synthetic state has 1 action class with high probability mass, while the remaining action (up) splits high probability mass between the first two instances. This is sufficient for the learners to distinguish all four actions, thus allowing for what Sucholutsky & Schonlau (2021a) call "less than one-shot learning". The 6 distilled MNIST digits in Figure 3a similarly distribute probability mass over 6 instances to train learners to distinguish the 10 classes of MNIST.

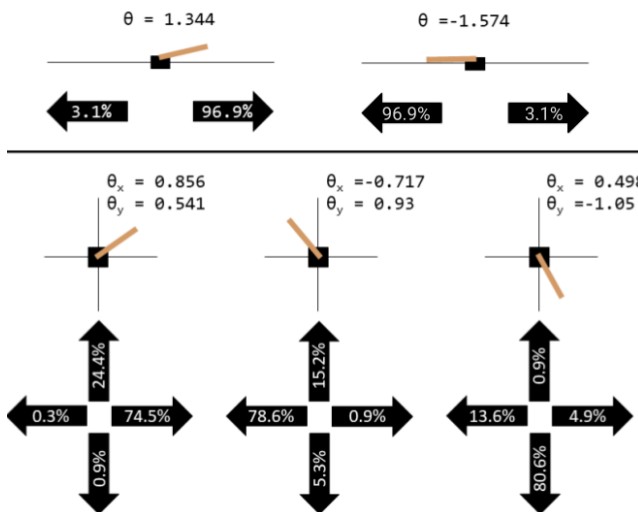

Figure 2: Minimum-sized distillations of the cart-pole problem with the standard one-dimensional cart-pole above and a 2-dimensional extension below. The soft action labels are softmaxed for improved readability. In all distilled instances, the cart's position and velocity are near zero; thus, the angular position of the pole is the only portion of the state visualized. In 1D cart-pole, the state and labels clearly show that the agent should move in the direction in which the pole is leaning. In the more complex 2D example, this is also shown, with one dimension prioritized over the others. In the right-most example, the label contradicts this in its horizontal labels, likely due to the priority of the vertical label due to the pole's vertical angle being much higher magnitude than its horizontal angle.

Focusing on the distilled data instances does not take into account how distillation meta-learning is performed. Distillation does not directly optimize data instances to resemble the original dataset or to mimic its features; rather, the distilled instances are optimized such that, when used in learning, they can cause a model to perform well on the original task. These distilled instances can be distorted beyond recognition. In addition, the instances are not learned independently, but as a single unit. In other words, the optimization structure of the original dataset—the cost surface—may be more informative about what distillation learns. Thus, in addition to interpreting dataset distillation as a process that produces compressions of the original dataset's features, we interpret it as a process that produces cost surfaces (also known as loss landscapes) that approximate features of the original dataset's cost surface, when computed on in-distribution learner models $\lambda \in \Lambda$. This perspective is vital for understanding how one-step learning on distilled datasets can achieve high performance on a variety of models.

## 3.2 Examining Distilled Cost Surface

The cost surface, or loss landscape, is a structure upon which optimization is performed, and thus provides the best view of an entire learning task. Though generally SGD is used to approximate the loss at a given set of parameters in training, the cost surface is a vital structure for interpreting learning by (stochastic) gradient descent optimization. The cost surface is constructed in $\mathbb{R}^{|\theta|}$, which is the parameter space of a given model or flattened neural network architecture. Each dimension represents a parameter and each point in the space represents a parameterization of the model. The value at that point is the sum cost over the entire dataset of the given parameterized model.

Due to the high dimensionality of parameter spaces in deep learning, visualizing cost surfaces in 2D necessitates the removal of information. Traditional cost surface visualization utilizes two axes, near-orthogonal in parameter space, with the cost represented by either a third axis or colored contours. A set of trained parameters, $\hat{\theta}$ is selected as the center point. Due to the high dimensionality of the parameter space, one can sample two random vectors $\delta, \eta \in \mathbb{R}^{|\theta|}$ for axes and can safely assume that the two vectors are near-orthogonal: $\delta^T \eta \approx 0$. Li et al. (2018) use this approach, but

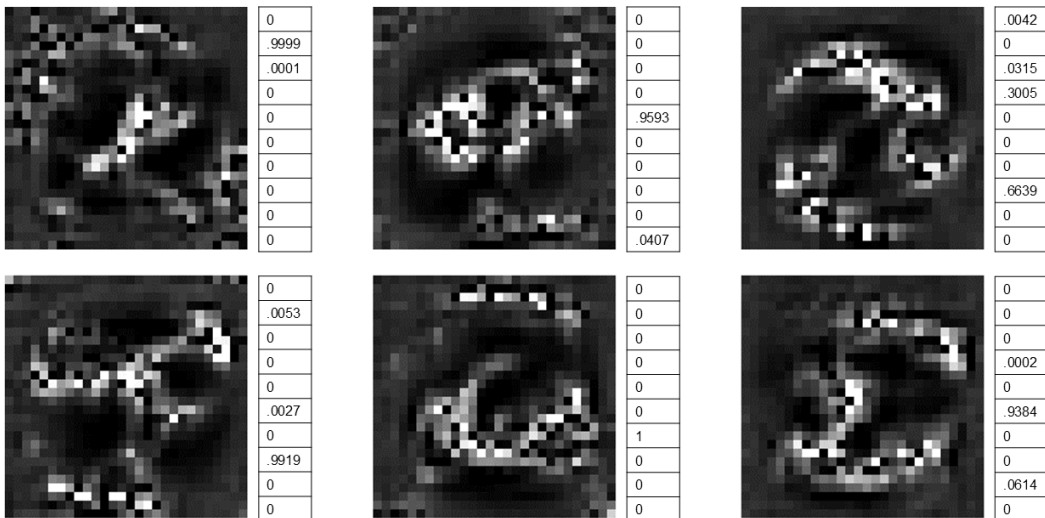

(a) A 6-instance distillation of MNIST. Each instance is shown as a monotone image with values clamped to [0,1]. To the right of each instance is the corresponding learned soft label vector, where the pixel values at each index represent the softmaxed probability of the class.

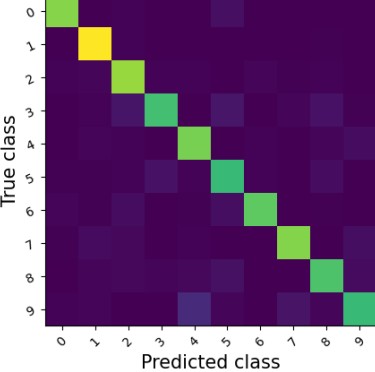

(b) MNIST validation confusion matrix for a model trained on the distillation in Figure 3a.

Figure 3: A minimum-sized distillation of MNIST (Figure 3a) and the validation confusion matrix of a model trained on the distillation (Figure 3b). For 10 classes, only 6 instances are needed due to soft classification labels (Sucholutsky & Schonlau, 2021a). The instances do not resemble true MNIST images, yet they have some interpretable features: the bottom-right image may have two fives superimposed; the top-right image has a 3-like shape but also has high probability mass on the 7 label. The confusion matrix demonstrates that, despite using only 6 training instances and having some classes with low overall probability mass, models can learn to distinguish all 10 classes.

add layerwise scaling. For each layer of the two vectors $v^l$ corresponding to $\hat{\theta}^l$, the vector is scaled $v^l := \frac{||\hat{\theta}^l||}{||v^l||} v^l$, where $|| \cdot ||$ is the Frobenius norm. The surface is created by sampling the loss at $\theta := \hat{\theta} + \alpha\delta + \beta\eta$, where $\alpha$ and $\beta$ determine the step size and range of the $\delta$ and $\eta$ axes, respectively. The plot is centered at $\hat{\theta}$, and the horizontal and vertical scaling represent $\alpha$ and $\beta$, respectively.

The cost of supervised learning tasks are easy to compute, but producing cost surfaces for reinforcement learning requires heuristics to estimate cost for a given network. RL environments are not represented by a single dataset, but rather, a dataset of experiences is produced as the learned policy interacts with the environment, often with stochastic elements. For our RL cost surfaces, we produce a single dataset from the policy represented by parameters $\hat{\theta}$ by running the policy repeatedly over the environment. We consider PPO's auxiliary critic network an integral part of the loss function, rather than representing its parameterization on the cost surface. While the dataset and critic we use

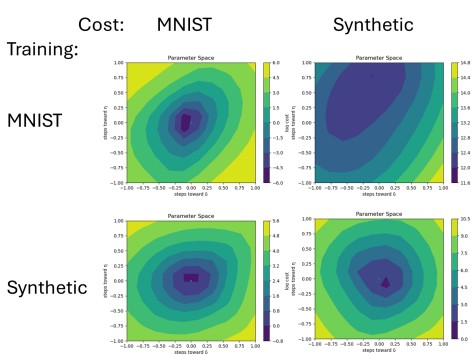
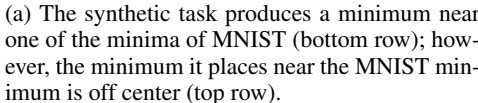
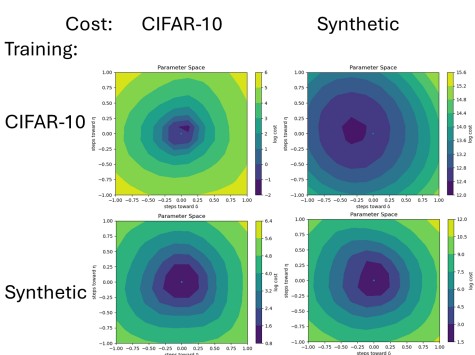

(a) The synthetic task produces a minimum near one of the minima of MNIST (bottom row); however, the minimum it places near the MNIST minimum is off center (top row).

(b) The surfaces appear similar to the MNIST surfaces in Figure 4a, though the CIFAR-10 distillation better models the location of the minima found through training directly on CIFAR-10.

Figure 4: A comparison between models trained directly on the original dataset (top) vs the corresponding synthetic dataset (bottom), visualized on the original (left) and synthetic (right) loss landscapes. The training (row) determines the position in parameter space (center point), while the source of the cost (column) determines the values on the surface. The axes $\delta$ and $\eta$ are randomly selected and scaled directions in parameter space, but are consistent in each subfigure.

are dependent on the policy represented by $\hat{\theta}$, we posit that this provides a decent approximation of the true cost surface of the RL learning task, especially as we examine the parameters close to $\hat{\theta}$.

Deep learning optimizes towards a minimum of the cost surface of a model on a learning task. A distilled dataset should therefore be constructed such that models initialized at any in-distribution point should approach a minimum of the true dataset's cost surface. We examine loss surfaces of models trained on distilled datasets to determine how distillation enforces this behavior.

We also posit that distillation can be used to cheaply produce approximations of a task's cost surface, due to the compressed nature of the distilled dataset. We suggest that this approximation can be more quickly searched for potential minima of the true cost surface, as the sum loss of some $\theta$ can be calculated in one forward pass of the network, rather than many. We demonstrate this by providing the time it took to produce visualizations of each task's surface (distilled and original tasks), when using our machine with an AMD Ryzen 7 2700X Eight-Core Processor CPU and a GeForce GTX 1080 Ti GPU. Other than batching data instances and using vectorized parallel RL environments, there was no explicit parallelization. Loss was computed on the GPU, all other operations including producing the figures was performed on one CPU core.

### 3.2.1 SUPERVISED LEARNING DISTILLATIONS

We examine the cost surfaces of models trained on supervised learning datasets (MNIST and CIFAR-10) and models trained on a corresponding distilled dataset. The results are shown in Figure 4. The synthetic surface places minima approximately around minima in the original dataset's surface. This leads the model to the targeted minima when trained on the synthetic dataset. The minimum found by optimizing on the original dataset is off-centered in both distilled tasks, which may explain the drop in performance of distillation-trained models. The distilled dataset attempts to approximate certain high-performing minima of the original dataset. Thus we can consider distillation training as a search problem: the original task's cost surface is searched by training the randomly initialized learners on the distilled dataset, and the distilled dataset is optimized to lead these learners to better points on the cost surface. It is reasonable that not all minima are well-represented on the synthetic cost surface, due to minima being missed in the search or due to limitations in the representational power of the limited-sized distilled dataset.

The cost surfaces produced with the costs of the distilled task are significantly cheaper to produce than those produced using the costs of the original task. It took approximately 13 minutes to produce an MNIST or CIFAR-10 surface with 100 sampled points, while the distilled surfacse took

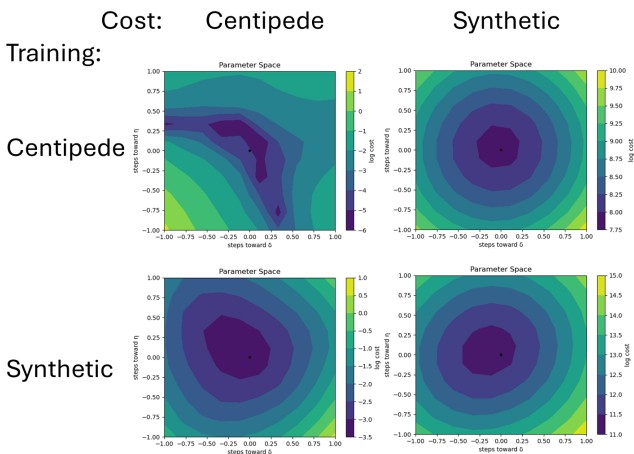

Figure 5: The loss landscapes of PPO on *Centipede* and a dataset distilled from *Centipede*. The distilled dataset performs minima matching well, though note that while the minimum found through RL is oblong on the RL landscape, the distilled set approximates it as a more regular ellipsoid.

approximately 0.2 seconds each. The true cost surfaces are expensive to produce, but the costs of approximating the true surface with a distillation is negligible. Thus, the distilled cost surface may be an ideal approximation when computation costs are a concern.

While this minima-matching scheme is a reasonable and intuitive way to model a cost surface, distillation can model cost surfaces without matching minima. So long as the distilled learners are trained for a fixed number of optimization steps, rather than to convergence, a cost surface could be constructed that sends a model's parameters from the initialization space to a real-task minimum without directly modeling that minimum. So long as the gradients map the initialization space points to minima, a synthetic dataset can approximate the real task.

### 3.2.2 CENTIPEDE COST SURFACES: GENERALIZATION TO OUT-OF-DISTRIBUTION ARCHITECTURES

We produce cost surfaces on the *Centipede* environment and a distillation of it (see Figure 5). Computation of the surfaces using *Centipede* PPO loss as cost took approximately 50 seconds, while the distillation cost surfaces took 0.3 seconds. The resulting plots closely match those of the supervised learning experiments, showing that distillation is providing a good approximation of the minima of the *Centipede* task, despite using a synthetic SL dataset to approximate the RL task.

Due to the cheap cost and fewer hyperparameters involved in producing the distilled cost surface, we propose the use of distillation to produce approximations of RL cost surfaces. Distillation eliminates auxiliary models from end-task training and sidesteps the issue of selecting which experience sets are valid for producing the cost surface, as well as removing randomness from the dataset.

Each architecture tested against a distilled dataset produces a different cost surface. While our distillation training used one training architecture per class, we demonstrate cost surfaces produced by a different architecture trained on our *Centipede* distillation in Figure 6. Since the minimum found by distillation training is a minimum on *Centipede*, we posit that the distillation is not overfitting to a single architecture, but rather generalizing. Thus, we should not think of distillation as producing one cost surface, but a distribution of cost surfaces over a distribution of models. This implies that the distillation is learning general-use knowledge, rather than simply placing minima on a small set of cost surfaces. The degree of generalization that distillation achieves is left for future work.

### 3.2.3 CART-POLE COST SURFACES: ALTERNATIVE TO MINIMUM-MATCHING

We produce cost surfaces on the cart-pole environment (approximately 15 seconds) and a cart-pole distillation (approximately 0.15 seconds). In Figure 7, we demonstrate that cart-pole distillation does not produce minima located at the RL cart-pole minima. The distillation is still successful,

 

(a) Centipede cost surface        (b) Distilled Centipede cost surface

Figure 6: Cost surfaces of an architecture trained on distillation for which it was out of the training architecture distribution. While the distillation is the same as that of Figure 5, the space is larger due to a different architecture being used. Note that the distilled dataset models a minimum for a cost surface that it has not seen, as this model was not used in producing the distillation.

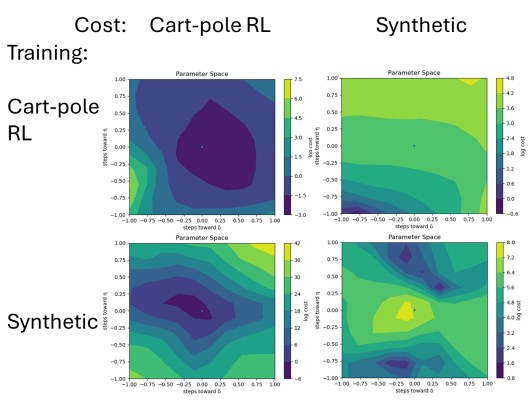 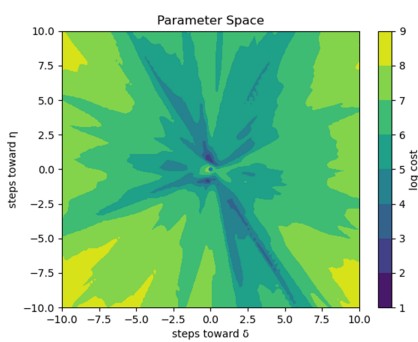

(a) Cartpole cost surfaces, varying task used to compute costs (columns) and trained center point location (rows)     (b) 10x zoomout of bottom-right landscape of Figure 7a

Figure 7: Visualization of cost surfaces of cart-pole and its distillation (Figure 7a) and a zoomed-out synthetic surface centered on a distillation-trained model's parameters. Note that both the RL agent and the agent trained on the distilled data reach minima on the cart-pole problem. Like with the supervised dataset distillations in Figure 4, the minimum found by the RL agent is not well-represented in the synthetic landscape. Unlike the supervised learning examples, the distilled model appears to be near a maximum on the synthetic landscape. This can be explained by zooming out on the landscape (see Figure 7b). On this larger scale, the maximum appears more like a minimum. Since the synthetic task is learned in a single SGD step, the model does not need to converge to a distillation-space minima, so long as it converges to a point near a minimum on the true task. If the gradient is directed to true minima for all points in the initialization space, the true minima do not need to be modeled by synthetic minima.

as Figure 8b demonstrates that one-step learning does not necessarily end at a synthetic minimum, though it ends at a minimum for RL cart-pole in Figure 7. Given that the distillation of *Centipede*, a more complex RL task, does perform minima matching, this may be due to how cart-pole reward is measured. Reward in *Centipede* is unbounded, while cart-pole is traditionally bounded—we utilized a maximum reward of 500. This bound produces large regions of high reward in the parameter space, with small policy changes yielding minimal changes in reward, while small policy changes in *Centipede* affect reward more drastically. This does not invalidate distillation's ability to approximate cost surfaces of other tasks—to verify whether minimum matching is occurring, one can produce a cost surface of the original task centered at the parameters found in distillation training to confirm if those parameters are at or near a local minimum of the original task.

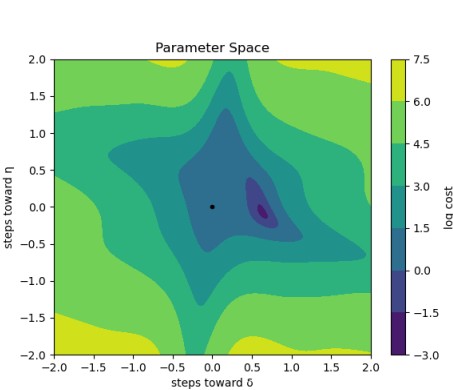

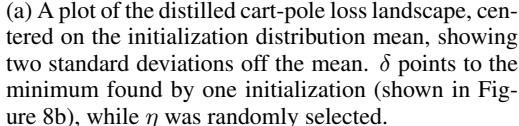

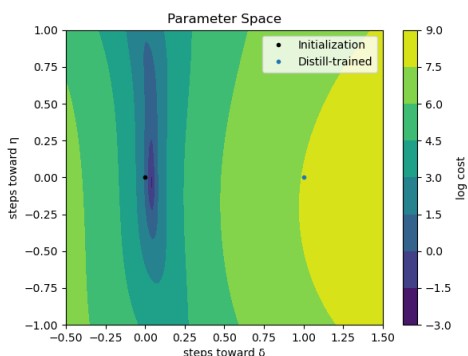

(a) A plot of the distilled cart-pole loss landscape, centered on the initialization distribution mean, showing two standard deviations off the mean. $\delta$ points to the minimum found by one initialization (shown in Figure 8b), while $\eta$ was randomly selected.

(b) The loss landscape over a one-step trajectory on distilled cart-pole. The gradient at the initialization points towards a minimum of cart-pole PPO, which it reaches in one step. Because one-step optimization is used, there is no need to place synthetic minima at the cart-pole minima, so long as the gradients in the initialization space point to cart-pole minima.

Figure 8: Distilled cost surfaces of cart-pole showing the model initialization space used in training the distillation and the learning trajectory of a model sampled from the distillation space.

## 4 CONCLUSION

We have examined dataset distillation by treating each distilled instance as an individual learning artifact and by treating the dataset as a singular structure and plotting cost surfaces. Examining individual distilled instances can reveal salient features for classes for image recognition and strategies to base policies around for interactive environments. The distribution of labels demonstrates how distillation can train a model to distinguish $c$ classes in fewer than $c$ instances.

By examining the cost surfaces, we have demonstrated that the distilled dataset approximates minima of the original dataset. This interpretation provides an intuitive explanation for the counter-intuitive level of approximation achievable with massive compression of training tasks through distillation. In addition, distillations can be used as cheap approximations of the true cost surface and can be calculated in just one forward pass per point on the surface. Because the cart-pole landscapes show that not all distillations will provide minimum-matching to approximate a landscape, testing losses on the original task around the distillation-trained point may be required to determine if minimum-matching is occurring.

This work focused on four deep learning benchmark tasks: MNIST, CIFAR-10, cart-pole, and Atari *Centipede*. Expanding to more complex tasks as future work would serve to further strengthen our arguments. We also focus only on the locations of minima of the cost surfaces; a deeper examination of the geometry and curvature of the space, as well as examination of other structures produced by model and dataset such as the model manifold may yield additional insights into the unreasonable effectiveness of one-step learning on distilled data.

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
