# OpenReview forum: "A Defense of One-Step Learning: Examining Single-Batch Distillations"
_ICLR.cc/2025/Conference — Submitted to ICLR 2025_

### Official Review · Reviewer_5y8o · 2024-10-31

**Soundness:** 1
**Presentation:** 2
**Contribution:** 2
**Rating:** 3
**Confidence:** 4

**Summary:**

This paper explores and tries to understand why extreme dataset distillation, where the number of samples is less than the number of classes and synthetically generated, can be used to train a model and the model can achieve comparable accuracy to a model trained on the full dataset, in the setting where one step learning is used.  This is explored in the MNIST, CIFAR10 supervised image classification task as a well as the cart-pole and Centipede reinforcement learning tasks. The paper suggests that the reason why extreme dataset distillation can perform well is that the models go to a local minima that is in the same space as the full datasets minima. To explore this they plot the loss landscapes of a model trained on the full dataset and the model trained on the distilled dataset, and then compare there loss landscapes made with the distilled and full dataset.  Therefore the overlap between these landscapes, explains the success of extreme dataset distillation and why it is to be expected.

**Strengths:**

1. Introduction is very clear, engaging and the problem is very well motivated, pleasure to read.

2. Contributions are layout clearly.

3. Two different modalities explored Vision and Reinforcement Learning.

4. The idea to explore why single batch distillations works is very interesting.

**Weaknesses:**

**Clarity**

In the abstract line 13-14: "Conventional wisdom suggests that single-step learning is not generalisable and should yield poor performance" I don't think this to be the case, as this paper [1] shows that stochastic training is not required for generalisation but if one is to use single-step learning, i.e. full-batch gradient decent, with a lot of explicit regularisation can achieve comparable accuracy to using SGD on CIFAR10.

In Section 2.2: DISTILLATIONS USED IN EXPERIMENTS, only the average result of training 1000 models is reported. Could you also report the standard deviation of these models? Also, why did you select 1000 instead of any other number? Why are the conventional trained models only trained once with the performance reported instead of averaged across 5 models? Why is train accuracy not reported?

**Qualitative Results:**

The comparison of the loss landscapes is poor as no metric is provided; even though the colours are the same, the values are not, making it hard to compare. To compare visually, ensure the images all use the exact colour mapping. With this, the idea of minima-matching needs to be adequately explained, other than stating the model achieves a low loss at/near the centre; however, these loss values are massively different. How are the loss landscapes the same or approximately matched?

**Mismatch between definition and result:**

Line 167-169: "While the CIFAR-10 distillation did not converge well, perhaps due to the model's risk of overfitting, the results demonstrate that even poor distillations function similarly to well-converged ones." There needs to be more evidence to support this claim; the MNIST case performs 14.1% worse than the entire dataset, suggesting that dataset distillation in this case will lead to poor-performing models regardless. I would go as far to say that this goes against the statement in the introduction line 036: "The distillation-trained model should perform comparably to the model trained on the original task." a 14.1% and 35.7% difference in the test accuracy is far from comparable accuracy. This is also the case for the reinforcement learning task with a large difference between Centipede of 1084 and 2D cart-pole: 134.1

**Figures**

Figure 2: 1-D cart pole. The directions are wrong on the second image; both images say 3.1% left and 96.9% right, irrespective of the direction of the 1-D cart pole. From the caption (Line 236:237: "In 1D cart-pole, the state and labels clearly show that the agent should move in the direction in which the pole is leaning.") it is my understanding that the second figure should have a bigger value on the left than on the right, as the pole is leaning left.

Figure 3a: The softmax probability of the classes is hard to read- can the values be reported instead of the colour gradient, which is hard to read?

Figure 4 needs to be explained clearly. It is difficult to tell which direction the trained (rows) and the datasets (columns). Could it be made more apparent?

Figure 8b) It is hard to tell the difference between the initialisation and the distil-trained; it appears that the distil-trained is on a higher part of the loss landscape and that the initialised model is closer to the minima.

Why are the CIFAR10 Single Batch dataset images excluded from the paper? I would have liked to have seen them at least added to the appendix.
z
**Minor points:**

Lines 48-49 need a citation; this is a bold statement that I would like verified.

Line 161: Starts with a "+"

Line 167: Please state the training accuracy instead of "(despite near-perfect training set accuracy)"


**Overall:**

The analysis and experimental setup is lacking, it is not made clear how comparisons are made other than visual inspection- which is warped due to having the colour maps the same even though the ranges are different. It is an interesting idea being explored, however the models do not achieve comparable accuracy, suggesting that the datasets are poor themselves. The hypothesis  does make sense that a good distilled dataset should result in a similar loss landscape to the full dataset, as it adequately captures the distribution of the data such that it is represented, however I do not think this is clearly explored or shown here especially as the loss is so high.


**References**

[1] Geiping, J., Goldblum, M., Pope, P.E., Moeller, M. and Goldstein, T., 2021. Stochastic training is not necessary for generalization. arXiv preprint arXiv:2109.14119.

**Questions:**

Please see the questions and comments in the weakness section with the following more concretely:

The idea of minima-matching needs to be adequately explained; I am unsure what you mean; could you better explain how the loss landscapes are compared? Is it a local or global comparison? Using a metric would make this objective; even something as simple as their absolute difference would be great.

Why, for the CIFAR10 dataset, do you use the same architecture as the MNIST, given that it does not produce good accuracy when trained conventionally? Why are not more common architectures such as a ResNet[1]  or VGG[2]?

I need clarification on why there is an attempt at re-framing loss landscapes to cost surfaces- could you elaborate more on why you chose to refer to the spaces as cost surfaces instead of loss landscapes?


[1] He K, Zhang X, Ren S, Sun J. Deep residual learning for image recognition. InProceedings of the IEEE conference on computer vision and pattern recognition 2016 (pp. 770-778).

[2] Simonyan K, Zisserman A. Very deep convolutional networks for large-scale image recognition. arXiv preprint arXiv:1409.1556. 2014 Sep 4.

---

> ### Comment · Reviewer_5y8o · 2024-11-26
>
> As it stands my rating of the paper will stay the same as the questions and weaknesses have not yet been addressed.

---

> ### Comment · Reviewer_5y8o · 2024-12-02
>
> I thank the authors for the changes to the PDF. However they have not address my core issues with the paper have not been sufficiently address. I will maintain my score.

---

> ### Author Response · Authors · 2024-12-02
>
> Thank you for your review. We apologize for our late response. We hope these points help clarify your concerns:
>
> **Weaknesses**
>
> Clarity:
>
> - While the provided reference supports the idea of one-step learning, we would argue that it also goes against conventional deep learning wisdom. We are not claiming that the conventional wisdom is correct; quite the opposite. In addition, we are not training on the full dataset, but a dataset several orders of magnitude smaller.
> - We considered 1000 runs to be sufficient to overcome the randomness of the random initialization and the randomness in each environment. For the RL experiments, we have also tested with 100 distillation-trained models over 100 episodes and reached similar results; thus, we assert that 1000 runs is sufficient. We agree that more runs with the original models would provide a fairer comparison.
>
> Qualitative Results:
>
> - We agree that metrics would provide clearer comparisons. While the loss values are different, they are not expected to be the same. Distillation must train a model in a single step of learning, thus there is no mechanism forcing the loss values to reflect those of the original dataset. Rather, we argue that distillation meta-learning creates minima in parameter space close to those of the original dataset.
>
> Mismatch between definition and result:
>
> - While the distillations do not reach the same performance as models trained on the original datasets, this learning was performed in a single step on less than one instance per class/action. In distillation, performance and compression are competing metrics. We agree that evaluating other distillations that prioritize performance over compression would strengthen our arguments.
>
> Figures:
>
> - Fig 2: Fixed
> - Fig 3a: Fixed
> - Fig 4: Noted, this could be made more clear visually. As stated in the the caption, the rows represent the training method (i.e. the centerpoint of the plot) and the columns represent the cost values used to produce the contours.
> - Fig 8b: Noted, the point colors are difficult to distinguish at the plot’s scale. You’re observation is correct, as noted in the caption, in this case the distillation does not converge to a minimum. This is a behavior possible with one-step learning, as the cost surface simply needs to map initializations to minima of the original task, not necessarily minima of the distilled task.
> - CIFAR10 surfaces are shown in Figure 4b.
> - This is not an extraordinary claim, the results are those reported in the paper and are the distillations used in the visualizations throughout. The 6-instance distillation of MNIST is shown in Figure 3.
> - Fixed
> - Agreed. We will provide the exact values for training accuracies, as well as standard deviations for all reported results.
>
> **Questions**
>
> - We agree that a metric would strengthen our examination of the minima. By minimum matching, we refer to the position of minima in parameter space, not necessarily the value or sharpness of the minima.
> - We will test with other architectures.
> - We believe the terms are equivalent.

---

> > ### Comment · Reviewer_5y8o · 2024-12-03
> >
> > ## Clarity 1
> > Conventional deep learning wisdom is a subjective term; here, it would depend on the literature known to the reviewer. To be more exact the text could be improved to be `it is a generally held practical belief that single-step learning is not generalisable and should yield poor performance`. This still considers the citation provided in the original review that shows full batch training can result in good-quality models. Still, it is not a practically viable option. At least to me, the result does not seem counter-intuitive; if the corset is a well-compressed form of the dataset, it would not surprise me that it could achieve comparable accuracy to the full dataset as a good compression would maintain a strong representation of the dataset with the fewest examples and thus full batch single step training could occur. But I can see how this would not be expected.
> >
> > ## Clarity 2
> >
> > Thank you for explaining.
> >
> > ## Qualitative Results
> >
> > The argument of `close in parameter space` could potentially be measured by controlling the size of the distilled dataset and comparing the distance between the model's layers in weight space as they are trained on the larger distilled datasets. To provide an understanding of how the size of the distilled dataset affects this closeness. One would expect that as more data is provided, the models are closer to the model trained on the full dataset in weight space. This would significantly strengthen your argument; however, as it stands, this is missing and thus does not hold up. I also argue what it means to be `close in parameter space` in such a high-dimensional space; many metrics would be required to make such a claim. For a visual inspection, providing a radial slice of the loss landscape, as done in [2], may aid this investigation, but metrics would be required.
> >
> > In addition, the loss landscapes are not that complex, which is probably due to the model's simplicity; exploring CIFAR10  with a VGG that demonstrates complex loss landscapes [4] could help further the perspective being claimed. Because it is hard to disentangle between the loss landscape being essentially bowl-like and the ease of the task, more complexity in the surface and your method showing similar troughs and peaks would add more validation behind the idea that it matches the loss surface.
> >
> > Although this cannot done within the time frame left, to strengthen this idea and add more support, it would be interesting to see if the two models can be connected and what the distance of connection is [1,3], are the minima in close to one another? Although an existing definition of close would be required to form the baseline.
> >
> > [1] Draxler, F., Veschgini, K., Salmhofer, M. and Hamprecht, F., 2018, July. Essentially no barriers in neural network energy landscape. In International conference on machine learning (pp. 1309-1318). PMLR.
> > [2] Fort, S., Hu, H. and Lakshminarayanan, B., 2019. Deep ensembles: A loss landscape perspective. arXiv preprint arXiv:1912.02757.
> > [3] Garipov, T., Izmailov, P., Podoprikhin, D., Vetrov, D. P., and Wilson, A. G. Loss Surfaces, Mode Connectivity, and Fast Ensembling of DNNs. ArXiv e-prints, February 2018. URL http://arxiv.org/abs/1802.10026.
> > [4] Li, H., Xu, Z., Taylor, G., Studer, C. and Goldstein, T., 2018. Visualising the loss landscape of neural nets. Advances in neural information processing systems, 31.
> >
> > ## Mismatch between definition and result:
> >
> > This is understood, however needs to be made clear in the paper. I do agree that adding additional experiments and evaluating with different data amounts would help strengthen the arguments.
> >
> > ## Figures
> >
> > Thank you for fixing the figures,
> >
> > As to fig 4: The caption does provide this context; however, it is not easy to process. As this is a big part of the story, it is essential to show which figure is more explicit. Making an explicit table with a more apparent heading would improve this.
> >
> > As to Fig 8b, I understand that the distillation case can result in a model not landing on a minima. Still, I am not sure what you mean by `as the cost surface simply needs to map initialisations to minima of the original task, not necessarily minima of the distilled task.` what do is meant by map here? This terminology is used without being fully explained.
> >
> > As to `CIFAR10 surfaces are shown in Figure 4b.` Sorry for the confusion; I meant the dataset examples, not the loss landscapes.
> >
> > As to `This is not an extraordinary claim...` Sorry, this was later realised with the text; I would add it as shown in this paper to clarify that it is based on your results and has not been shown before.

---

> > > ### Comment · Reviewer_5y8o · 2024-12-03
> > >
> > > ## Questions
> > >
> > > 1. It would be essential to show how your method compares against using a single batch of training data instead of the full set, given that one claim is that it allows for quicker production of the loss landscapes. I am also unsure what you mean by `minimum matching` The minima are not equivalent. How are they matched? When doing loss landscapes, point zero, zero is the current model's loss; it then explores the landscape from that point by perturbing the weights in two randomly orthogonal directions to give an overview of the general landscape. Therefore the model's current minima/position will be at zero, zero. How can one say they match?
> > >
> > > 2. Great, this will significantly improve the paper.
> > >
> > > 3. In future work, it is recommended to use the language of the literature and stay consistent with it. Otherwise, it requires the reader to hold both equivalent terms in their head instead of one. Sticking with loss landscapes is recommended, as this is more frequently used in literature.
> > > ## Overall
> > > Thank you for taking the time to respond. I, however, will not change my score. The paper is qualitative, and the minima matching terminology must be better explained. The responses and paper changes have yet to address my core concerns surrounding using metrics, additional architectures, figure scale matching, etc.

---

> > > > ### Author Response · Authors · 2024-12-03
> > > >
> > > > We understand you keeping your original score. Thank you for your responses; we are grateful for your feedback which will help us strengthen our research and our paper.

---

### Official Review · Reviewer_7JzF · 2024-11-03

**Soundness:** 1
**Presentation:** 2
**Contribution:** 2
**Rating:** 3
**Confidence:** 5

**Summary:**

This paper studies the performance of dataset distillation under the regime of one gradient update under the meta-model matching framework. The study shows that a single step gradient update distilled data can achieve decent task performance, and the distilled dataset having ideal properties with respect to the real data with similar data features and similar loss landscapes towards a specific local minima.

**Strengths:**

1. Since dataset distillation as a field right now is mostly empirical, the inner-workings on why it works is understudied. This work studies the loss landscape of the distilled data, which can provide important insights for better design of dataset distillation algorithms.
2. The study on RL datasets cart-pole and centipede is interesting since most of dataset distillation works focuses on image recognition.
3. The paper demonstrate that distilling with one gradient step can achieve decent task performance on CIFAR-10 with accuracy of 28% using fewer than 1 image per class through soft-labeling.

**Weaknesses:**

1. It is unclear whether findings in this paper will translate to other distillation algorithms, and therefore, how it will fit in the existing body of research. While the paper demonstrates the ability to achieve decent performance with one-step gradient update, the performance is very subpar compared to other distillation algorithms such as BPTT [1], which achieves 49% with 10 examples, or Trajectory Matching [2], which achieves 46% with 10 examples.
2. The method useful to justify why a solution is a local minima is not sound. Visualization through random vectors projection was originally designed to capture the non-convexity of the loss landscape and is insufficient for the understanding the optimization trajectory (section 7.1 of [3]). To better understand the optimization trajectory, visualization with PCA direction proposed in section 7.2 of [3] can be used. To quantitatively justify local minimum, one would have to reason about the sharpness (second derivative/Hessian) of the loss landscape [4].

[1] Deng, Zhiwei, and Olga Russakovsky. "Remember the past: Distilling datasets into addressable memories for neural networks." Advances in Neural Information Processing Systems 35 (2022): 34391-34404.

[2] Cazenavette, George, et al. "Dataset distillation by matching training trajectories." Proceedings of the IEEE/CVF Conference on Computer Vision and Pattern Recognition. 2022.

[3] Li, Hao, et al. "Visualizing the loss landscape of neural nets." Advances in neural information processing systems 31 (2018).

[4] Yao, Zhewei, et al. "Pyhessian: Neural networks through the lens of the hessian." 2020 IEEE international conference on big data (Big data). IEEE, 2020.

**Questions:**

1. Curious towards whether one-step learning generate datasets with different properties compared to multi-step learning approaches such as BPTT. Existing work shown that data distilled with popular algorithm seems to capture early trajectories rather than specific local minimum [1]. Curious whether authors have any insights on whether single-step learning changes this property?
2. One potential issue with visualization through random projections is that every solution will look like a minima with sufficiently large step size. What does the loss landscape look like for CIFAR-10 with smaller step sizes?

[1] Yang, William, et al. "What is Dataset Distillation Learning?." Forty-first International Conference on Machine Learning.

---

> ### Author Response · Authors · 2024-12-02
>
> Thank you for your review. We apologize for our late rebuttal, but we hope that we can clarify a few points:
>
> - W1: We agree that testing on other distillation methods would strengthen our results.
> - W2: This is a valuable point. We will apply your proposed method to our experiments and compare to our current results.
> - Q1: We have not compared the two approaches directly. In our reinforcement learning experiments, the distillation clearly learns fine on later trajectories. If this were not the case, the model would be learning on data gathered only with poor RL policies and would not be able to reach the high performance seen in the RL experiments.
> - Q2: We used the scaling of the magnitude of the trained network layers for all plots except where clearly stated.

---

### Official Review · Reviewer_XZV1 · 2024-11-03

**Soundness:** 2
**Presentation:** 2
**Contribution:** 2
**Rating:** 5
**Confidence:** 3

**Summary:**

This work explores how dataset distillation enables models to achieve effective one-shot learning from reinforcement learning perspective. Despite conventional belief that single-step learning would not generalize well and would perform poorly, distilled datasets allow models to closely approximate the results of direct-task training across a wide range of model architectures. The authors analyze both the distilled data instances and the cost surfaces they generate, finding that the distilled datasets not only replicate features of the original data but also shape cost surfaces that guide models from their initial states into local minima of the true task’s cost surface.

**Strengths:**

1. The paper provides an in-depth examination of cost surfaces generated by distilled datasets, offering a valuable perspective on how dataset distillation guides models to local minima. By analyzing the distilled instances, the authors demonstrate that the compressed data retains critical task-relevant features, which helps improve interpretability, especially for simpler tasks.

2. To the best of the knowledge, this is the first paper that investigates the dataset distillation in reinforcement learning scenario.

**Weaknesses:**

- While the cost surfaces generated by distillations show promising results, the paper does not fully address potential scalability issues when applied to larger models or datasets, which could present computational challenges. I suggest the authors provide more theoretical verification for the claim.
- The study relies on one method of distillation, but an evaluation of alternative distillation methods could provide a broader understanding of how different techniques impact cost surfaces and model performance. Methods like DATM, SDC, IDC should also be considered as baselines for further exploration.
[1] Guo Z, Wang K, Cazenavette G, et al. Towards lossless dataset distillation via difficulty-aligned trajectory matching[J]. arXiv preprint arXiv:2310.05773, 2023.
[2] Wang S, Yang Y, Wang Q, et al. Not all samples should be utilized equally: Towards understanding and improving dataset distillation[J]. arXiv preprint arXiv:2408.12483, 2024.
[3] Kim J H, Kim J, Oh S J, et al. Dataset condensation via efficient synthetic-data parameterization[C]//International Conference on Machine Learning. PMLR, 2022: 11102-11118.
- More complex datasets and benchmarks should also be considered. The experiments mainly focus on relatively simple datasets, such as MNIST and CIFAR-10, as well as cart-pole and Centipede environments. The paper could be strengthened by evaluating the method on more complex tasks, such as high-resolution image datasets or NLP tasks, to assess its scalability.
- This paper does not use the correct ICLR template. Specifically, there is no ''Under review as a conference paper at ICLR 2025'' note in the paper.
- Minor: some notations are wrong. For example, in figure 1, the authors write T0, T_0, and also TO, which generate mistakes. I suggest the authors polish the notations and representation. In line 161, there is a unexpected + symbol.

**Questions:**

1. How does the proposed approach handle high-dimensional data, such as images with complex structures or datasets with numerous features? It would be beneficial to understand how the technique performs in such scenarios.
2. Given the success in various supervised and reinforcement learning tasks, could this method be extended to other domains, like natural language processing or time-series analysis?

---

> ### Author Response · Authors · 2024-12-02
>
> Thank you for your review. We apologize for our late response, yet we hope we can respond to a few of your concerns:
>
> - W1: While we do not prove that distillation itself is scalable, that is well beyond the scope of this paper. The goal of the paper was to understand one-step learning, not necessarily prove one-step learning's usefulness. Our visualization method is scalable to distillations, as distillations are significantly smaller than the original tasks. We do admit that visualizing large datasets is computationally expensive using the method we applied, making the comparisons difficult. Yet, if our assertions about how the distilled surface mimics minima placement of the original surface, this would provide a clear use-case for distillation - examining complex cost surfaces with little computational cost.
> - W2/3: We agree that testing our method on more distillation methods and tasks would strengthen our results.
> - W4: Fixed
> - W5: Fixed
> - Q1: The most complex task we tested on was Atari Centipede.
> - Q2: In theory, it could expand to any domain so long as there is a differentiable loss function.

---

### Official Review · Reviewer_N5Yq · 2024-11-03

**Soundness:** 2
**Presentation:** 3
**Contribution:** 2
**Rating:** 3
**Confidence:** 4

**Summary:**

This article explores the effectiveness of one-step learning by analyzing distilled data instances and cost surfaces. The experiments demonstrate that the distilled dataset not only replicates the characteristics of the real dataset but also generates suitable cost surfaces, enabling one-step training to guide the model from the initialization space to a local minimum of the actual task cost surface.

**Strengths:**

1. The experiments encompass a variety of benchmark tasks, including image recognition and reinforcement learning.
2. The article presents a task-agnostic distillation algorithm (Algorithm 1) and thoroughly details the various steps, parameter settings, and selection of loss functions relevant to different tasks in the distillation process. This clarity aids readers in understanding and replicating the experiments.

**Weaknesses:**

1. **Theoretical Depth**: While the effectiveness of the distilled dataset is demonstrated experimentally, the theoretical framework explaining why an appropriate cost surface emerges during the distillation process is somewhat lacking. The conclusions largely rely on empirical observations. The article primarily documents experimental details and phenomena, missing an in-depth analysis that could better inform the design and application of the methods discussed.
2. **Generality of Experimental Validation**: The authors have chosen algorithms and datasets that are too simplistic and limited. The method selected by the authors targets distillation with single-step training, as they mentioned, single-step learning is the expected outcome for distilling these types of datasets, but what about distillation methods for other categories? There are many new related meta-learning algorithms, as well as many methods outside of meta-learning (e.g., trajectory matching).

**Questions:**

1. The observations of the cost surface and loss curvature have also been observed, analyzed, and used as a basis for proposing improvement methods in the work of others[1]. They conducted a more detailed analysis of loss curvature. Have you combined your analysis with theirs?
2. The current experiments are too simplistic. To my knowledge, there are several advanced algorithms for dataset distillation that are not very computationally expensive, and these experiments are entirely affordable. Can the authors conduct experiments on more recent works and larger datasets?

[1] Shin, Seungjae, et al. "Loss-curvature matching for dataset selection and condensation.”

---

> ### Comment · Reviewer_N5Yq · 2024-11-26
>
> I continue to uphold my evaluation, as the authors remained uninvolved in the rebuttal phase, leaving my concerns unresolved.

---

> ### Author Response · Authors · 2024-12-02
>
> Thank you for your review. We apologize for our late responses. Your concerns cannot be fully addressed in the short time allotted for the discussion period. As you suggested, we will prioritize testing with other distillation methods. We will also focus on providing stronger theoretical backing for our work. We were unaware of the work you cited; thank you for bringing it to our attention. We will examine it and combine our analysis with theirs.

---

### Official Review · Reviewer_CKhM · 2024-11-04

**Soundness:** 2
**Presentation:** 2
**Contribution:** 2
**Rating:** 5
**Confidence:** 4

**Summary:**

This paper presents a study that inspects the loss surfaces of distilled datasets. The authors show the results on various tasks, including Atari Centipede and standard image benchmarks.

**Strengths:**

+ The paper is easy to follow and understand
+ The authors performed relatively though coverage on different domains.

**Weaknesses:**

- I'm not sure what the paper is contributing. The distilled datasets can train a model that performs similarly, the results itself hint the loss surface probably shares similar local minima.

- A lot of related works are missing. [1] analyzes the DD task in depth. [2] made multi-step BPTT work in DD. One easy question to ask and study further is, one-step BPTT (used in this paper) underperforms multi-step BPTT[2], what has been changed in the loss landscape when number of steps is increased?

[1] What is Dataset Distillation Learning? icml'24
[2] Remember the Past: Distilling Datasets into Addressable Memories for Neural Networks. neurips'22

**Questions:**

See above. I'm not sure what this paper is contributing and the current version seems to be not sufficient. Giving deeper understanding of DD is interesting and the authors should consider further complete the work to submit to another future venue.

---

> ### Comment · Reviewer_CKhM · 2024-11-27
> **Final review**
>
> I'll maintain my rating.

---

> ### Author Response · Authors · 2024-12-02
>
> Thank you for your review. We apologize for our late response. We will examine the works you've provided and add them to the related works and examine where we might utilize their methods to strengthen our papers.
>
> As far as the contribution, we believe the results we have achieved with one-step learning distillations and minimum-sized distillations are counterintuitive. Thus, our work attempts to find out how a model is capable of learning in one gradient descent step. We believe that prior works in distillation, which focus on interpreting distilled dataset instances, are missing the full picture. We assert that the full picture can only be seen by examining distillation as a learning task through the cost surface, rather than as disconnected instances in a dataset.

---

### Author Response · Authors · 2024-11-26

We thank the reviewers for their feedback and apologize for our late response. Due to time and resource limitations, we have set our priorities elsewhere, but with the extended discussion period, we will endeavor to respond to the questions and concerns of the reviewers as best we can.

Since the first deadline is the paper PDF revision, we will focus on the revision before responding to individual reviewers. Note that some of the reviewers' concerns require significant time and resources to implement. We will attempt to rectify all the concerns that we can in this limited time, but we are likely not able to implement all the feedback at the moment. We will take this feedback into account for future revisions of the paper as needed, and we are grateful for the reviewers for their efforts in helping us improve our research.

---

> ### Author Response · Authors · 2024-11-27
> **Paper Revision Changes**
>
> We have implemented the following changes to the manuscript:
>
> - Changed Figure 1 notation
> - Corrected Figure 2 labels
> - Replaced Figure 3 labels w/ text labels
> - Minor formatting issues fixed: header corrected, minor edits to text

---

### Meta-Review · Area_Chair_v73d · 2024-12-20

**Metareview:**

This paper examines how distilled datasets can enable one-shot learning by analyzing the distilled data and the cost surface of the distilled dataset. The authors found that the distilled data not only mimics the features of the real dataset but also helps models reach the local minima of the real dataset with one-shot learning.

Reviewers’ comments highlight several strengths of the paper. All reviewers agree that the study is comprehensive, as it covers multiple domains. Reviewers (XZV1, 7zJF, N5Yq) commend the paper for its in-depth examination of the cost surface produced by the distilled dataset. Reviewers (CKhm, 5y8o) praise the readability of the paper.

However, reviewers also raised several concerns. These include the lack of discussion on related works (Reviewer CKhM), questions about the validity of the contribution (Reviewer CKhM, 7zJF), the absence of a theoretical explanation for the distillation process (Reviewer N5Yq, XZV1), insufficient experiments on larger datasets, benchmarks, and alternative distillation algorithms (Reviewer XZV1, N5Yq, 7zJF), issues with the method's soundness in explaining why a solution is a local minimum (Reviewer XZV1, 5y8o), and a lack of clear presentation (Reviewer XZV1, 5y8o).

During the rebuttal phase, the authors acknowledged the weaknesses pointed out by the reviewers and attempted to address their concerns. However, many of these issues remain unresolved.

After careful consideration of all factors, the AC recommends rejecting the paper.

**Additional Comments On Reviewer Discussion:**

During the rebuttal, reviewer 5y8o questioned the assertion that single-step learning performs poorly, the unclear presentation, and insufficient evidence for some claims. Although the concern regarding the number of runs was addressed, other concerns by reviewer 5y8o had not yet been addressed. Moreover, the concerns of the reviewers (CKhm,N5Yq, XZV1) were not fully addressed, such as the absence of a theoretical explanation, the validity of the contribution, and so on. Given that most of the reviewers' concerns were not resolved, AC recommends rejecting the paper.

---

### Decision · Program_Chairs · 2025-01-22

Reject